# Right Question is Already Half the Answer: Fully Unsupervised LLM Reasoning Incentivization

**Qingyang Zhang** *
Tianjin University

**Haitao Wu**
Tianjin University

**Changqing Zhang** †
Tianjin University

**Peilin Zhao**
Shanghai Jiao Tong University & Tencent AI Lab

**Yatao Bian** ‡
National University of Singapore & Tencent AI Lab

## Abstract

Existing methods to enhance the reasoning capability of large language models predominantly rely on supervised fine-tuning (SFT) followed by reinforcement learning (RL) on reasoning-specific data. These approaches critically depend on external supervisions–such as labeled reasoning traces, verified golden answers, or pre-trained reward models. In this work, we propose Entropy Minimized Policy Optimization (`EMPO`), which makes an early attempt at fully unsupervised LLM reasoning incentivization. By minimizing the semantic entropy of LLMs on unlabeled questions, `EMPO` achieves competitive performance compared to supervised counterparts. Specifically, without any external supervision, `EMPO` boosts the accuracy of Qwen2.5-Math-7B Base from 33.7% to 51.6% on math benchmarks and improves the accuracy of Qwen2.5-7B Base from 32.1% to 50.1% on MMLU-Pro. Primary analysis are also provided to interpret the effectiveness of `EMPO`. Code is publicly available at https://github.com/QingyangZhang/EMPO.

## 1 Introduction

Large language models (LLMs) have demonstrated exceptional potential in challenging tasks such as mathematical reasoning [1, 2, 3] and code generation [4]. A prevailing paradigm for training reasoning LLMs involves firstly performing supervised fine-tuning (SFT) and then reinforcement learning (RL), or iterative combinations of both, applied to reasoning-specific datasets after pretraining [5]. Unfortunately, these methods typically depend on large-scale reasoning datasets with various forms of supervised information, such as human-labeled reasoning traces, verified golden answers, or an additional pre-trained reward model. As a consequence, endowing LLMs with powerful reasoning capability through human

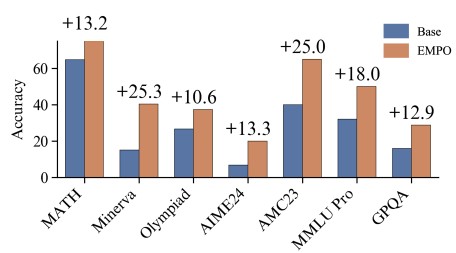

Figure 1: Improvement of the proposed method on Qwen2.5-7B and Qwen2.5-7B-Math model.

---

*Work done during an internship at Tencent AI Lab.

†Correspondence to Changqing Zhang <zhangchangqing@tju.edu.cn>

‡Yatao Bian is the project lead.

39th Conference on Neural Information Processing Systems (NeurIPS 2025).

experts is becoming increasingly time-consuming and costly, which greatly limits the scalability and broader adoption of reasoning models.

To mitigate this, previous work employs self-consistency to construct pseudo data and deploy supervised finetuning for better performance [6]. However, the performance improvement is limited and under risks of model collapse [7]. Recent advancements, such as the pioneering work PFPO [8], frame the labeling of solutions as evaluation against test cases and then leverage self-consistency to generate pseudo test cases. Despite the promising results, the proposed method still necessitates supervision from instruction finetuning data and supervision signals from the frontier LLMs to initialize the RL process. Another more recent work [9] introduces a two-stage framework to construct self-rewarding reasoning models using self-generated data followed by RL. Despite the superior performance, the proposed method relies on a ground-truth verifier to obtain self-correction reasoning traces by rejection sampling. These approaches inspire our exploration of a critical open question: **How can we incentivize LLM reasoning capacities in a fully unsupervised manner?**

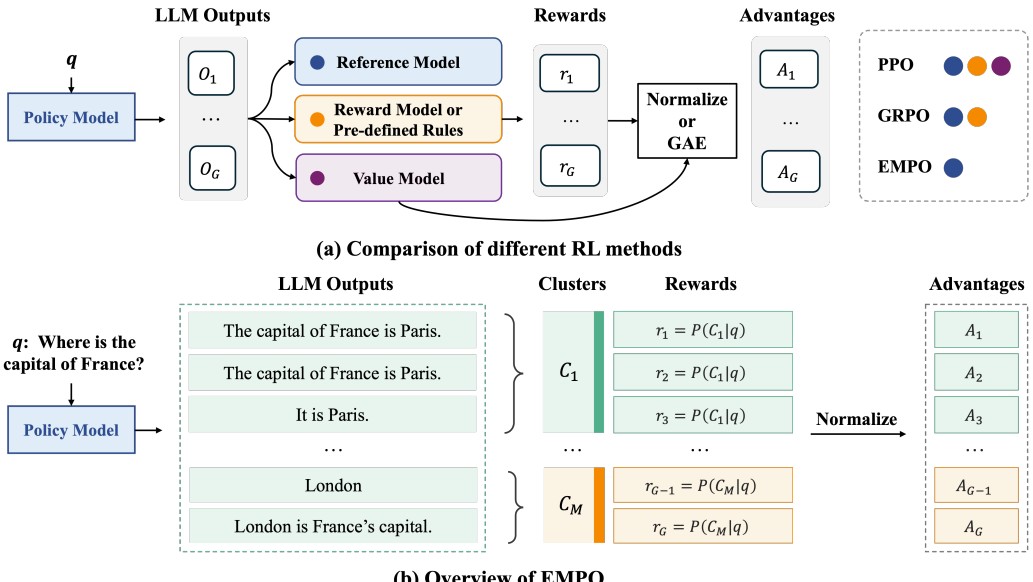

(a) Comparison of different RL methods

(b) Overview of EMPO

Figure 2: Overview of the proposed method. (a) Previous method like PPO [10] or GRPO [11] typically relies on external supervised signals, e.g., a pretrained reward model or golden answers. (b) The proposed Entropy Minimized Policy Optimization (EMPO) samples a set of responses from the current policy model, and then builds semantic clusters according to their equivalence. By continuously minimizing the entropy at a meaning level, our method achieves competitive benchmark performance without any external supervision, i.e., rule-based reward, pre-defined test cases or an pre-trained reward model.

Recent advanced DeepSeek-R1-Zero [12] demonstrates robust reasoning capabilities without dependency on SFT data. By directly initiating RL from the base model, DeepSeek-R1-Zero autonomously evolves sophisticated reasoning behaviors such as reflection and self-critic by exploring the reward signals provided by rule-based rewards. i.e., verified golden answers or an additional pre-trained reward model. Inspired by the success of DeepSeek-R1-Zero, our motivation is to devise a fully unsupervised approach for powerful reasoning capability. Specifically, we propose a novel reinforcement learning algorithm termed as Entropy Minimized Policy Optimization (EMPO), which incentivizes the reasoning capability of LLMs in a fully unsupervised manner by minimizing their predictive entropy in a latent semantic space. This method optimizes the model to favor reasoning traces yielding consistent answers, enhancing output reliability. The semantic entropy objective we propose to minimize is a well-established measurement of LLMs' uncertainty, which extends beyond mathematical reasoning to free-form question-answering tasks. We further introduce entropy thresholding to filter unreliable reasoning traces, stabilizing the unsupervised training process. Experiments on various

tasks including mathematical reasoning and free-form natural reasoning are conducted to validate the proposed method. Our contributions are summarized as follows:

- We propose an effective and principled strategy called Entropy-Minimized Policy Optimization (EMPO) for incentivizing the reasoning capabilities of LLMs in a fully unsupervised manner.

- We establish semantic entropy as a potent intrinsic reward signal for guiding LLM reasoning. Our analysis confirms a strong negative correlation between semantic entropy and model accuracy, validating its efficacy as a robust, unsupervised optimization objective that drives models towards generating more consistent and reliable outputs.

- Experiments on both math reasoning tasks with deterministic golden answers and free-form natural reasoning tasks are conducted to validate the efficacy and versatility of EMPO. Additionally, we provide critical insights into EMPO's mechanism, demonstrating that its effectiveness stems from an enhanced ability to consistently select and prioritize strong, pre-existing reasoning pathways learned during pre-training, rather than instilling fundamentally new reasoning skills. This underscores EMPO's strength in efficiently eliciting and refining latent capabilities within base models.

## 2 Related Work

**Self-Supervised and Semi-Supervised Reasoning.** To address the dependency on labeled data, several self-supervised and unsupervised methods have emerged. Huang et al. [6] propose a self-improvement framework where LLMs generate high-confidence answers using Chain-of-Thought (CoT) prompting and self-consistency, subsequently fine-tuning on these pseudo-labels. However, the performance gains are often limited, and there is a risk of model collapse, as noted in [7]. Recently, Patel et al. [13] apply self-improvement to web navigation tasks in WebArena, fine-tuning on synthetic data generated by the model itself. Li et al. [14] enhance long-context reasoning via SeaLong, sampling multiple outputs and optimizing with Minimum Bayes Risk. These methods, while reducing reliance on external labels, still involve supervised fine-tuning steps, contrasting with EMPO's fully unsupervised RL approach. A concurrent work, i.e., test-time reinforcement learning (TTRL) [15] directly obtains pseudo label by major voting and then conducts RL on test prompts at inference time, whereas our EMPO strictly maintains the separation between training and testing phases for ensuring that the model remains unexposed to any test prompts during training. Furthermore, while TTRL is currently limited to mathematical tasks, our approach is applicable to more general free-form reasoning tasks.

**Self-Rewarding and RL-based Reasoning.** RL has become a prominent technique for enhancing LLM reasoning, often leveraging external or self-generated rewards. Yuan et al. [16] propose using the LLM itself via LLM-as-a-Judge prompting to provide rewards during training, reducing reliance on human feedback. Similarly, recent works [9, 17] incorporate the training of model's self-verification capability into RLVR process, which improve the reasoning and verification capabilities at the same time. PFPO [8] frame solution labeling as evaluation against test cases, yet still rely on instruction fine-tuning and frontier LLM signals for RL initialization. ETPO [18] augments RL with an entropy bonus to promote exploration, differing from EMPO's entropy minimization focus. DeepSeek-R1[12] demonstrate robust reasoning via RL from a base model, using rule-based rewards. Seed-GRPO [19] leverages semantic entropy for reward shaping, which enables dynamic adjustment of policy update magnitudes based on question uncertainty. These methods highlight a spectrum of supervision levels, positioning EMPO as unique in its fully unsupervised nature, leveraging semantic entropy as an internal reward.

**Entropy Minimization and Semantic Consistency.** Entropy minimization is a well-established technique in semi-supervised and unsupervised learning, with roots in traditional machine learning. Many previous work [20, 21, 22] has demonstrated that minimizing entropy on unlabeled data can improve classification accuracy by encouraging model confidence. Test-time adaptation methods like Tent adapt models to new domains by minimizing entropy on test data, filling domain gaps without additional labels. These approaches highlight the potential of entropy minimization as an unsupervised objective, which EMPO leverages for LLM reasoning by extending it to semantic entropy

[23] in a latent space. More recent work [24] compute the Semantic Entropy to quantify the diversity of the modelâĂŹs high-level strategic plans, reinforcing its relevance.

## 3 Method

We propose an RL-based method to minimize the entropy of LLM generations in a latent semantic space for incentivizing its reasoning capability. We term our method Entropy-Minimized Policy Optimization (EMPO), which is devised in a fully unsupervised manner without any forms of external supervised information.

### 3.1 Preliminaries

Recent advancements in reinforcement learning have demonstrated remarkable breakthroughs in enhancing the reasoning capabilities of LLMs. Taking the representative RL technique Group Relative Policy Optimization (GRPO) [11] used by DeepSeek-R1-Zero [12] as an example. GRPO first samples a group of outputs $\{o_1, \cdots, o_G\}$ from the policy model $\pi_\theta$ and then optimizes it by maximizing the following objective:

$$\mathcal{J}_{\text{GRPO}} = \mathbb{E}_{[q \sim P(Q), \{o_i\}_{i=1} \sim \pi_{\theta(O|q)}]}$$
$$\left[ \frac{1}{G} \sum_{i=1}^{G} (\min(A_i, \text{clip}(1, 1 - \epsilon, 1 + \epsilon)A_i) - \beta \ KL(\pi_\theta||\pi_{ref})) \right], \quad (1)$$

where $\beta$ is a hyper-parameter which avoids the policy model to diverge too far away from the reference model $\pi_{ref}$. $\epsilon$ clips extreme advantages for stability. $G$ is the number of samples in one group. $A_i$ is the advantage computed by normalizing the rewards within each group as $A_i = \frac{r_i - mean(\{r_1, \cdots, r_G\})}{std(r_1, \cdots, r_G)}$. In math reasoning task, the reward can be computed by predefined rules:

$$r_i = \begin{cases} 1 & \text{if verifier}(o_i, a) = \text{True} \\ 0 & \text{otherwise} \end{cases}, \quad (2)$$

where a verifier is used to determine the correctness of $o_i$ by comparing it with the golden answer $a$.

Unlike the above example, we consider fully unsupervised optimization settings where there are no golden answers to verify the correctness of model predictions. In this circumstance, we only have unlabeled reasoning problems $P(Q)$. Such problems were freely raised by users during the deployment of LLMs. Given a pre-training LLM $\pi_\theta$ parameterized by $\theta$, our goal is to enhance its reasoning ability by only utilizing the unlabeled user problems $\{q_i\}_{i=1}^n$, which requests minimized cost of data collection.

### 3.2 Semantic Entropy Minimization Objective

Entropy is a classical unsupervised objective in the traditional semi-supervised and unsupervised learning fields [20, 25]. Previous works in computer vision show that by continuously minimizing the entropy on unlabeled samples after pre-training, the classification accuracy of machine learning models can be significantly improved to fill the domain gaps [21, 22]. The basic intuition behind entropy minimization is that a robust model should not only fit labeled data well but also make confident and consistent predictions on unlabeled data. This principle encourages the model to avoid ambiguity and make decisive predictions, thereby enhances generalization. In this work, we choose semantic entropy [23] as our unsupervised optimization objective, which is a natural extension of classical Shannon entropy specified for large language models. Intuitively speaking, minimizing semantic entropy encourages the LLMs' outputs to be more consistent in semantic level rather than format, and thus the final answers are expected to be more reliable.

Specifically, semantic entropy first samples a group of outputs $\{o_1, \cdots, o_G\}$ and then clusters the output sequences according to their meaning. That is, if two outputs share the same meaning (i.e., they are bidirectionally entailed), they should be merged into one same cluster in the semantic space. This can be done without notable computational cost by predefined rules such as N-gram, regular expressions or an additional small language model. Once built such a set of meaning clusters $\{c\}$ in

semantic space, we then approximate the probability over the meanings as the proportion of sampled answers as

$$p(c_j|x) \approx |c_j|/G, \tag{3}$$

where $c_j \in \{c\}$ is the $j$-th meaning cluster. $|c_j|$ denotes the numbers of outputs that belong to $c_j$. Finally, given question $q$, the semantic entropy (denoted as $H$) over the model's output meanings distribution can be estimated as follows

$$H = - \sum_{c_j \in \{c\}} p(c_j|q) \log p(c_j|q). \tag{4}$$

As proven by previous work, semantic entropy has a strong negative relationship with model accuracy, which can be used as an efficient measurement to detect unreliable LLM generations such as confabulation and hallucination [23, 26, 27]. Motivated by this, we propose to leverage semantic entropy as an unsupervised optimization objective for incentivizing the reasoning capability of LLM.

### 3.3 Entropy-Minimized Policy Optimization

We propose Entropy-Minimized Policy Optimization (EMPO), an RL-based method that optimizes the pre-trained large language model $\pi_\theta$ to favor low semantic entropy responses given unlabeled user questions $\{q_i\}_{i=1}^n$. Given input questions, EMPO incentivizes the outputs that belong to higher probability meaning cluster, and thus minimizes the semantic entropy over the meaning distribution. Specifically, given a question $q$, our EMPO first samples a group of output $\{o_1, \ldots, o_G\}$ from the current model $\pi_\theta$ and then merges them into a set of $M$ meaning clusters $\{c_1, \ldots c_M\}$. As we mentioned before, this can be done without notable computational cost (please refer to the quantitative results in Appendix F) by predefined rules such as N-gram, regular expressions or an additional small language model (SLM)[4]. Once built such a meaning set, EMPO approximately minimizes the semantic entropy $H$ by maximizing the following objective

$$\mathcal{J}_{\text{EMPO}} = \mathbb{E}_{[\{q\} \sim P(Q), \{o_i\}_{i=1}^G \sim \pi_\theta(O|q)]} \frac{1}{|G|} \sum_{i=1}^{|G|} (A_i), \; A_i = \frac{r_i - mean(\{r_1, \cdots, r_G\})}{std(r_1, \cdots, r_G)} \tag{5}$$

where $A_i$ is the advantage of output $o_i$ calculated by normalizing the rewards. Unlike GRPO in which the rewards is calculated depending on external supervision such as pre-defined rules or an reward model, in EMPO, the reward assigned for the $i$-th outputs $o_i$ is the likelihood of its meaning cluster, i.e.,

$$r_i = p(c_j|q), \text{ where } l(o_i) = c_j, \tag{6}$$

where the meaning likelihood $p(c_j|q)$ is approximated by Eq. 3. Intuitively, the outputs convey higher-probability meanings are of higher advantages, and are therefore incentivized through training.

**How to Mitigate Potential Reward Hacking?** Note that different from verifiable rule-based reward, which inherently resists reward hacking risks, optimizing unsupervised entropy objectives may permit trivial solutions. For instance, models could exploit the reward signal by overfitting to high-confident but wrong predictions for the most frequent semantic clusters without carefully reasoning process. To address this, we implement a straightforward entropy thresholding strategy, restricting optimization to prompts exhibiting moderate uncertainty via dual threshold criteria. Specifically, two entropy thresholdings are deployed to filter out user queries $q$ that result in overly high or low entropy unreliable answers. Extremely high entropy indicates that the model is highly uncertain, and thus its predictions are prone to be unreliable. In addition, continuously optimizing on responses with already low entropy is redundant and at the risk of overconfidence [28]. The final optimization objective of EMPO is

$$\mathcal{J}_{\text{EMPO}} = \mathbb{E}_{[\{q\} \sim P(Q), \{o_i\}_{i=1}^G \sim \pi_\theta(O|q)]}$$

$$\left[ \frac{1}{|G|} \sum_{i=1}^{|G|} (\min(A_i, \text{clip}(1, 1-\epsilon, 1+\epsilon)A_i)) \right], \tag{7}$$

$$\text{s.t. } \delta_{low} < H < \delta_{high}$$

---

[4]Such a SLM does not provide explicit or direct supervision signals regarding the correctness or quality of reasoning for a given query. The "unsupervised" nature of EMPO refers to its independence from labeled (query, correct-answer) pairs or (query, valid-reasoning-trajectory) pairs for learning the reasoning task itself. More discussions are in Appendix J.

where $H$ is the semantic entropy defined in Eq. 4. The questions results in highly unreliable answers with entropy greater than $\delta_{high}$ are filtered out. Besides, we also filter out low-entropy answers to maintain the diversity of model outputs and further avoid potential reward hacking. Following previous work [29], we remove the KL constraint for better performance. $\epsilon$ clips extremely high or low advantages for stability similar to common practice.

## 4 Experiments

### 4.1 Experimental Settings

We conduct experiments on multiple datasets including both closed-form math reasoning tasks and free-form natural reasoning tasks. Our `EMPO` shows competitive performance by purely RL in a fully unsupervised manner compared to supervised finetuning and RL methods.

**Prompt Collection and Data Engineering.** For mathematical reasoning, following the common practice [30, 8, 31], we adopt 20,000 prompts randomly selected from NuminaMath-CoT dataset [32] for training[5] without additional data engineering. For free-form natural reasoning tasks, we adopt the prompts from Natural Reasoning[6], a large-scale dataset consisting of diverse reasoning questions from multiple domains (e.g., Physics, Computer Science, Economics, Social Sciences and more). For training efficiency, we filter out the questions with over-long prompt or reference answer. Besides, taking inspiration from [33, 34, 35], we use the response length of Llama3.3-70B-Instruct as a difficulty estimation metric, and filter out samples with response lengths exceeding 4096 tokens. The remaining samples are simpler for stabilizing the training process. The final training subset is consisted of 18,000 questions. More details can be found in Appendix G.

**Evaluation.** ∘ For mathematical reasoning, the performance is evaluated on a diverse suite of benchmarks including Minerva Math, MATH, AMC23, OlympaidBench and AIME24. ∘ For free-form natural reasoning, we evaluate on MMLU-Pro [36] and GPQA [37] benchmarks, which consist of challenging reasoning-focused problems across various subjects, e.g., biology, business, chemistry, computer science and so on. We prompt the model to reason step by step and output the final answer within "\boxed{}" and report the multi-choice accuracy. Without specific clarification, all evaluations are conducted using zero-shot prompting and greedy-decoding.

**Model training.** ∘ For mathematical reasoning tasks, we train Qwen2.5-Math-1.5B and 7B Base models with our `EMPO`. The baselines we consider include supervised finetuning (SFT), online direct preference optimization (ODPO) [31] and the representative GRPO. We also compared with Qwen2.5-Math Instruction models for a more comprehensive comparison, where the instruction model is trained by iteratively supervised finetuning and RL on private data. ∘ For free-form natural reasoning tasks, we initialize from Qwen2.5-3B, 7B and 14B Base models. Different from mathematical reasoning, it is difficult to adopt rule-based reward for free-form question-answering tasks without deterministic golden answers. We consider the corresponding Instruct model, the Base model with or without few-shot CoT prompt as baselines. Besides, we also compare with SFT where the Base model is tuned to fit the response of Llama3.3-70B-Instruct. For more results on other model families beyond the Qwen2.5 series (e.g., Llama3), please refer to the Appendix D.

- SFT: We train models by supervised finetuning via Open-Instruct [38] with a fixed learning rate of $1 \times 10^{-6}$, a global batch size of 128 and train for 1 epoch with a max length of 2048.
- GRPO: We implement GRPO via verl [39]. We sample 16 and 12 responses for each prompt for mathematical and natural reasoning tasks respectively. For Qwen2.5-Math model series, we train the model for 300 steps with a maximum generation length of 3096. For OctoThinker model series, we train the model for 100 steps with a maximum generation length of 16K. We adopt a train prompt batch size of 256 and mini-batch size of 32. Specifically, we adopt clipping by importance ratio based on sequence likelihood proposed by GSPO [40] for stable off-policy RL training. Following [41], we only use the rule-based accuracy reward and do not adopt format-reward. The accuracy reward is implemented as follows: If the response contains the correct final answer within "\boxed{}", it receives a reward of $1.0$. If the model prediction is wrong, it receives a reward of $0.0$.

---

[5]https://huggingface.co/datasets/RLHFlow/numia_prompt_dpo1
[6]https://huggingface.co/datasets/facebook/natural_reasoning

- Online-DPO: Recent advanced Online-DPO first samples a set of responses and then verifies and selects the responses with highest reward and lowest reward as a preference pair. We directly copy the results from [31], where the model is trained for 7 iterations. Each iteration involves 2 training epochs and 20K training samples, i.e., 140K training samples in total.
- `EMPO`: Most hyper-parameters of our method, e.g., number of generations, max generation length, batch size, learning rate are the same with GRPO. In mathematical reasoning tasks, we use a set of regular expressions to merge the outputs into meaning clusters. For more general free-form natural reasoning, we leverage General-Verifier[7] (a compact small language model with 1.5B parameters) to determine whether two outputs are of the same meaning or not following [23, 26]. A concrete example can be found in Appendix C. Specifically, if the final predictions (i.e., the contents within "\boxed{}") of two model outputs are bidirectionally implicating, then we merge them into one semantic cluster ignoring their reasoning traces. More details are in Appendix E.

## 4.2 Main Results

### 4.2.1 Performance on Mathematical Reasoning Tasks.

We conduct experiments on mathematical tasks to evaluate our method. The main results are shown in Table 1. `EMPO` has successfully incentivized the Qwen2.5-Math Base model with reasoning capability without dependency on any external supervision. We observe a substantial improvement in the average performance on commonly used mathematical reasoning benchmarks from 33.0% to 42.6% and 33.7% to 51.6% on Qwen2.5-Math-1.5B and 7B models, respectively. Notably, through fully unsupervised RL training, the 1.5B and 7B model has both achieved competitive performance (42.6% and 51.6%) near to Qwen2.5-Math-Instruct (42.8% and 51.7%), where the latter depends on private dataset and multi-stage iteratively supervised fine-tuning and reinforcement learning.

Table 1: Accuracy on mathematical reasoning benchmarks. We report the pass@1 accuracy tested with greedy decoding. The results of ODPO are directly copied from [31]. Here $q, r, a$ denote the dependency on questions, human-verified reasoning traces and golden answers respectively.

| | Supervision | MATH | Minerva Math | Olympiad Bench | AIME24 | AMC23 | Avg. |
|---|---|---|---|---|---|---|---|
| *frontier model* | | | | | | | |
| Llama-3.1-70B-Instruct | $\{q, r, a\}$ | 64.6 | 35.3 | 31.9 | 16.7 | 30.1 | 35.7 |
| Eurus-2-7B-PRIME | $\{q, r, a\}$ | 79.2 | 38.6 | 42.1 | 26.7 | 57.8 | 48.9 |
| *1.5B model* | | | | | | | |
| Qwen2.5-Math | None | 66.4 | 19.1 | 33.8 | 3.3 | 42.5 | 33.0 |
| Qwen2.5-Math-Instruct | $\{q, r, a\}$ | 75.2 | 33.8 | 42.8 | 6.7 | 52.5 | 42.2 |
| Qwen2.5-Math w/SFT | $\{q, r, a\}$ | 61.8 | 26.1 | 27.1 | 3.3 | 37.5 | 31.2 |
| Qwen2.5-Math w/Rand | $\{q, r, a\}$ | 65.0 | 26.1 | 30.7 | 10.0 | 55.0 | 37.4 |
| Qwen2.5-Math w/GRPO | $\{q, a\}$ | 78.0 | 37.1 | 39.1 | 10.0 | 50.0 | 42.8 |
| Qwen2.5-Math w/EMPO | $\{q\}$ | 77.6 | 36.0 | 39.5 | 10.0 | 50.0 | 42.6 |
| *7B model* | | | | | | | |
| Qwen2.5-Math | None | 70.2 | 12.5 | 30.8 | 10.0 | 45.0 | 33.7 |
| Qwen2.5-Math Instruct | $\{q, r, a\}$ | 80.8 | 41.9 | 49.2 | 13.3 | 67.5 | 50.5 |
| Qwen2.5-Math w/SFT | $\{q, r, a\}$ | 72.2 | 34.6 | 33.2 | 10.0 | 45.0 | 39.0 |
| Qwen2.5-Math w/Rand | $\{q, r, a\}$ | 73.0 | 26.5 | 37.0 | 26.7 | 52.5 | 43.1 |
| Qwen2.5-Math w/ODPO | $\{q, a\}$ | 76.8 | 30.9 | 37.9 | 26.7 | 62.5 | 47.0 |
| Qwen2.5-Math w/GRPO | $\{q, a\}$ | 82.4 | 45.2 | 47.6 | 23.3 | 60.0 | 51.7 |
| Qwen2.5-Math w/EMPO | $\{q\}$ | 81.4 | 42.3 | 46.1 | 23.3 | 65.0 | 51.6 |

### 4.2.2 Performance on Natural Free-form Reasoning Tasks.

We present the results on free-form natural reasoning tasks in Table 2. On the MMLU-Pro benchmark, our `EMPO` improves the accuracy from 32.1% to 50.1% and 32.7% to 58.8% on Qwen2.5-7B and 14B Base model respectively. Besides, on more challenging GPQA benchmark, `EMPO` results in increasing accuracy from 15.9% to 28.8% on 7B model, 30.6% to 35.3% on 14B model. Notably, we observe that the SFT baseline fails to consistently improve model performance. We hypothesize that this is

---

[7]https://huggingface.co/TIGER-Lab/general-verifier

due to the noise in the reference responses within the Natural Reasoning training data (as mentioned by [33]). This phenomenon further underscores the practical potential of our proposed method.

Table 2: Accurascy results on free-form natural reasoning benchmarks. We report pass@1 accuracy tested with greedy decoding. Here $\{q, r, a\}$ denote the dependency on questions, human-verfied reasoning traces and verifiable golden answers respectively.

| | Supervision | STEM | MMLU Pro | | | | GPQA |
| | | | Humanities | Social | Other | Avg. | |
|---|---|---|---|---|---|---|---|
| *3B model* | | | | | | | |
| Qwen2.5-Base | - | 8.32 | 5.35 | 7.42 | 4.15 | 6.83 | 11.2 |
| Qwen2.5-Base 5-shot | $\{q, r, a\}$ | 34.7 | 26.2 | 47.9 | 35.9 | 35.3 | 13.8 |
| Qwen2.5-Instruct | $\{q, r, a\}$ | 44.8 | 30.7 | 56.0 | 47.1 | 44.5 | 28.2 |
| Qwen2.5-Base w/SFT | $\{q, r, a\}$ | 19.8 | 10.4 | 28.0 | 18.4 | 19.1 | 11.5 |
| Qwen2.5-Base w/Rand | $\{q, a\}$ | 17.0 | 29.8 | 49.6 | 28.4 | 26.0 | 18.5 |
| Qwen2.5-Base w/GRPO | $\{q, a\}$ | 32.2 | 27.7 | 49.8 | 38.7 | 35.2 | 17.1 |
| Qwen2.5-Base w/EMPO | $\{q\}$ | 31.7 | 26.2 | 48.1 | 36.7 | 34.1 | 20.6 |
| *7B model* | | | | | | | |
| Qwen2.5-Base | - | 30.1 | 23.8 | 45.9 | 34.3 | 32.1 | 15.9 |
| Qwen2.5-Base 5-shot | $\{q, r, a\}$ | 45.7 | 36.3 | 59.1 | 49.4 | 46.8 | 23.5 |
| Qwen2.5-Instruct | $\{q, r, a\}$ | 56.9 | 38.1 | 64.1 | 58.6 | 55.2 | 35.3 |
| Qwen2.5-Base w/SFT | $\{q, r, a\}$ | 32.6 | 7.1 | 15.8 | 30.1 | 25.6 | 22.4 |
| Qwen2.5-Base w/Rand | $\{q, r, a\}$ | 45.8 | 30.7 | 60.4 | 50.9 | 46.4 | 25.3 |
| Qwen2.5-Base w/GRPO | $\{q, a\}$ | 57.1 | 36.2 | 64.4 | 56.6 | 54.5 | 33.8 |
| Qwen2.5-Base w/EMPO | $\{q\}$ | 52.4 | 34.6 | 59.0 | 50.9 | 50.1 | 28.8 |
| *14B model* | | | | | | | |
| Qwen2.5-Base | - | 30.8 | 28.0 | 44.4 | 33.0 | 32.7 | 30.6 |
| Qwen2.5-Base 5-shot | $\{q, r, a\}$ | 51.9 | 35.8 | 63.4 | 54.4 | 51.4 | 33.2 |
| Qwen2.5-Instruct | $\{q, r, a\}$ | 63.6 | 47.1 | 73.8 | 66.7 | 62.9 | 42.9 |
| Qwen2.5-Base w/SFT | $\{q, r, a\}$ | 37.0 | 27.8 | 40.2 | 38.0 | 36.1 | 28.5 |
| Qwen2.5-Base w/GRPO | $\{q, a\}$ | 62.9 | 42.1 | 68.6 | 59.8 | 59.6 | 35.6 |
| Qwen2.5-Base w/EMPO | $\{q\}$ | 61.4 | 41.6 | 68.3 | 60.0 | 58.8 | 35.3 |

#### 4.2.3 Training Dynamics

We further conduct experiments to investigate the reliability of our unsupervised reward signals. As shown in Figure 3, the unsupervised reward signals of `EMPO` have a strongly negative correlation with the true rewards based on golden answers. Thus, by continuously minimizing the semantic entropy objective, the model can boost its accuracy in a fully unsupervised manner. Furthermore, in our experiments, we observed that during the training process, the response length of Qwen2.5-Math gradually shortened before stabilizing within a certain range. In contrast, the inference length of Qwen2.5 continuously increased. However, this variation in length did not show a clear correlation with performance improvement.

#### 4.2.4 Additional Results on Broader Model Family beyond Qwen2.5

To further validate the effectiveness of `EMPO` on broader model family beyond Qwen2.5-Math, we conduct additional experiments on OctoThinker-3B-Long-Base [42], which built upon mid-training starting from the Llama-3 model family. The results are shown in table 3. We sample 16 responses for each prompt and train the model for 100 steps with batch size of 256 and mini-batch size of 32. The maximum generation length is 16K. EMPO successfully improve the average accuracy from 7.7% to 27.6%, which demonstrates its effectiveness on various model family.

## 5 Discussion and Conclusion: The Role of Unsupervised Learning in Eliciting Pre-Trained Reasoning Capabilities

The strong empirical performance of `EMPO`, particularly its ability as a fully unsupervised method to match or even slightly outperform supervised counterparts like GRPO (as observed with the 7B model), prompts a deeper examination of how such reasoning incentivization mechanisms work. This is especially pertinent given the counterintuitive observation that these substantial improvements on

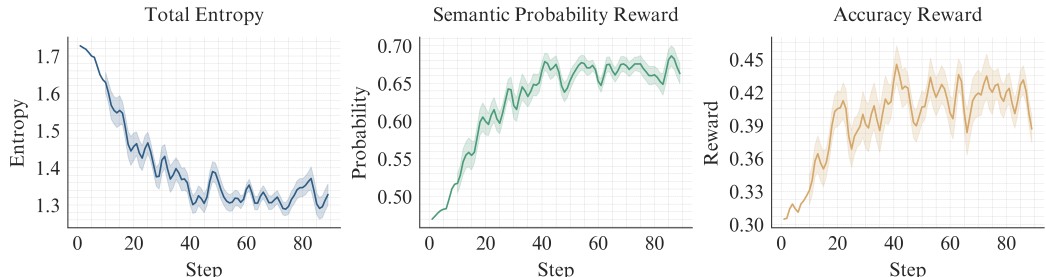

Figure 3: We visualize the training dynamics when tuning Qwen2.5-Math-7B Base model with `EMPO` on 20K prompts randomly sampled from NuminaMath-CoT. The left illustrates the running average of semantic entropy (Eq. 4). The middle shows the trend of our unsupervised reward as defined by Eq. 6. The right shows the model accuracy on training data at each RL steps. Along the unsupervised RL-based training trajectory, `EMPO` establishes a stable learning process with consistently decreased semantic entropy and improved accuracy.

Table 3: Additional results on broader model family beyond Qwen, i.e., OctoThinker-Long-3B.

| | Supervision | MATH | Minerva Math | Olympiad Bench | AIME24 | AMC23 | Avg. |
|---|---|---|---|---|---|---|---|
| *3B model* | | | | | | | |
| OctoThinker Base | None | 15.8 | 2.9 | 7.5 | 0.0 | 12.5 | 7.7 |
| OctoThinker-Zero | $\{q, a\}$ | 69.6 | 27.6 | 32.0 | 13.3 | 42.5 | 37.0 |
| OctoThinker w/GRPO | $\{q, a\}$ | 65.0 | 23.5 | 27.3 | 6.7 | 32.5 | 31.0 |
| OctoThinker w/EMPO | $\{q\}$ | 60.6 | 17.3 | 23.6 | 6.7 | 30.0 | 27.6 |

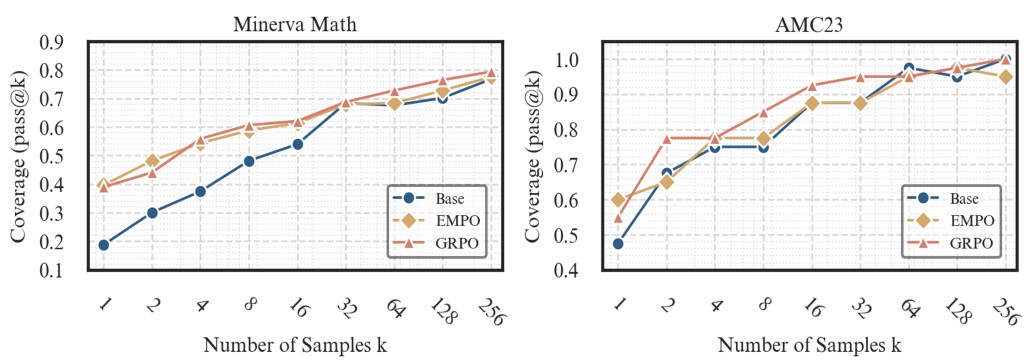

Figure 4: Pass@k curves of Qwen2.5-Math-7B Base model and its counterparts trained with GRPO and our `EMPO` on Minerva Math and OMNI reasoning benchmarks. Pass@k measures the probability that at least 1 of the top $k$ generated solutions is correct. Pass@1 is equivalent to accuracy, as it checks if the single solution is correct. When $k$ is small, RL-trained models outperform the original base model. However, as k increases (e.g., into the tens or hundreds), the performance of the base models often converges with, or even exceeds, that of the RL-trained models.

benchmarks are achieved without a consistent increase in response length or clear evidence of an "Aha moment" – a hypothesized sudden emergence of enhanced reasoning capabilities.

To dissect the nature of the improvements conferred by reinforcement learning (RL) post-training, we investigated its influence on pass@k accuracy. This metric is crucial as recent studies [43, 44] suggest that RL may not fundamentally expand the inherent reasoning capacities of LLMs beyond those already embedded in their pre-trained base. As depicted in Figure 4, our findings align with this perspective. Both GRPO and `EMPO` significantly enhance pass@k scores for small to moderate values of k (e.g., k=16 or 32) compared to the base model. This demonstrates an improved efficiency in surfacing correct reasoning paths with fewer attempts. However, as k becomes substantially large,

the performance of these RL-trained models tends to converge with, and is sometimes surpassed by, that of the base model.

This convergence at high k values, coupled with our qualitative observations that the base models themselves already exhibit sophisticated reasoning behaviors such as pausing, self-correction, and backtracking (see Appendix for examples), strongly indicates that the foundational reasoning pathways are largely pre-existing. Consequently, RL post-training, whether supervised or unsupervised like EMPO, appears to primarily refine the model's ability to efficiently access, prioritize, and consistently select these latent reasoning patterns, rather than instilling fundamentally novel ones. The observed improvements in pass@1 (accuracy) are thus likely a consequence of this enhanced sampling efficiency.

These empirical insights from the pass@k analysis lend considerable support to the emerging consensus that pre-training shoulders the primary burden of endowing LLMs with their core abilities. We align our interpretation with prior insights from [45]: "*Pretraining does all the hard work. One big bet is that the pretraining phase grants all the abilities to the base LM, and finetuning is simply like a style transfer which positions the model to the right output space.*" Under this conjecture (or more precisely, an emerging, but not yet unanimously accepted consensus [43]), we attribute the efficacy of our method to the robust pretraining process of the Qwen2.5 Base model: If a base model possesses strong inherent reasoning capabilities, the subsequent challenge is not necessarily to teach it new reasoning skills from scratch, but rather to effectively elicit and guide these existing skills.

*EMPO's success highlights that intrinsic reward signals, derived purely from the model's objective to minimize semantic entropy and thus achieve greater consistency in its outputs, can be surprisingly potent for this elicitation process.* In a well-pre-trained model, outputs that are semantically consistent are more likely to align with correct and coherent reasoning. EMPO leverages this by incentivizing the model to favor such consistent outputs, effectively guiding it to refine its selection from its collection of existing reasoning strategies without requiring external validation of correctness.

In conclusion, while RL techniques, including EMPO, may not be forging entirely new fundamental reasoning capabilities beyond what pre-training provides, their role in significantly enhancing the sampling efficiency and reliability of accessing these pre-trained abilities is of paramount practical importance. Optimizing models for such efficiency is crucial for real-world applications. EMPO, by achieving this through a fully unsupervised framework, offers a particularly scalable, cost-effective, and practical approach to unlocking and refining the vast reasoning potential embedded within pre-trained LLMs, especially in domains where curated supervisory data is scarce or prohibitively expensive to obtain.

# 6 Limitations

This manuscript presents a new training algorithm that improves the reasoning capability by minimizing semantic entropy. By encouraging semantically coherent outputs, the model's reasoning capability can be boosted without any external supervision. There exists several noteworthy limitation of EMPO. The unsupervised nature of EMPO removes the dependency of external supervision at the cost of potential reward hacking risks. The model may be highly overconfident on its prediction regardless correctness. Besides, similar to other RL for reasoning methods, EMPO may exhibit undesirable behaviors that stem from pretraining corpus of the Base model, e.g., hallucination, bias and so on. Entropy minimization may not only increase accuracy by encouraging consistency, but also make the bias more severe. Thus, more studies are necessitated to mitigate such issues for large reasoning models.

# 7 Acknowledgement

This work is supported by the National Natural Science Foundation of China (62376193 and 61925602). YB is supported by the National University of Singapore SoC (grant no: A-0010308-00-00). We thank Ma Huan for the insightful discussion regarding the role of RL in incentivization the reasoning capabilities of models and writing. We also thank Zongbo Han (BUPT), Chenqian Gao (MBZUAI), Qichao Wang (NTU) and Hengyu Liu (CUHK) for their helpful discussion about the project and comments on the manuscript.

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

# Appendices

## A  Prompt Templates

We provide the prompt templates used for training and evaluation.

For mathematical reasoning tasks, we adopt the following reasoning prompt template similar to Online-DPO-R1 project [31] for both training and testing. During testing, we found that by adding system prompt, the accuracy of Qwen2.5-Math Base model can be better on mathematical benchmarks. However, system prompt would not help in natural reasoning tasks. Thus we use the same test prompt (start with system prompt) for both Base model and finetuned models in mathematical tasks. In natural reasoning tasks, we do not add system prompt for Base models.

---
**Mathematical Reasoning Training and Evaluation Template**

<|im_start|> system
Please reason step by step, and output your final answer within \boxed{}.
<|im_end|>
<|im_start|>user
{Question} Let's think step by step and output the final answer within \boxed{}.
<|im_end|>
<|im_start|>assistant

---

To train models with our `EMPO` for free-form natural reasoning tasks, we adopt the following reasoning prompt template similar to that we used in mathematical tasks for training.

---
**Free-form Natural Reasoning Training Template**

<|im_start|>system
Reason step by step, and output your final answer within \boxed{}.
<|im_end|>
<|im_start|>user
{Question} Reason step by step and output the final answer within \boxed{}.
<|im_end|>
<|im_start|>assistant

---

Since the MMLU-Pro and GPQA are both close-formed multi-choice benchmark. To evaluate the natural reasoning capability of the models, we use the following prompt template during testing.

---

**MMLU Pro Test Template for Base Models**

Question: {Question} Reason step by step and output the final answer (the correct letter choice from A-P) within \boxed{}.
Answer:

---

**Few Shot MMLU Pro Test Template**

Question: {Question in Demonstration 1} Reason step by step and output the final answer (the correct letter choice from A-P) within \boxed{}.
Answer: Let's reason step by step. CoT of Demonstration 1 Therefore, the correct answer is Answer of Demonstration 1.
...
(Omit more demonstrations for readability)
...
Question: {Question} Reason step by step and output the final answer (the correct letter choice from A-P) within \boxed{}.
Answer:

---

**MMLU Pro Test Template for Finetuned Models (SFT and RL)**

<|im_start|>system
Reason step by step, and output your final answer (the correct letter choice from A-P) within \boxed{}.
<|im_end|>
<|im_start|>user
{Question} Reason step by step and output the final answer (the correct letter choice from A-P) within \boxed{}.
<|im_end|>
<|im_start|>assistant

---

**GPQA Test Prompt for Base Models**

Question: {Question} Reason step by step and output the final answer (A, B, C, or D) within \boxed{}.
Answer:

---

**Few Shot GPQA Test Prompt for Base Models**

Question: {Question in Demonstration 1} Reason step by step and output the final answer (A, B, C, or D) within \boxed{}.
Answer: Let's reason step by step. CoT in Demonstration 1 Therefore, the correct final answer is Answer in Demonstration 1.
...
(Omit more demonstration for readability)
...
Question: {Question} Reason step by step and output the final answer (A, B, C, or D) within \boxed{}.
Answer:

<|im_start|>system
Reason step by step, and output your final answer (A, B, C, or D) within \boxed{}.
<|im_end|>
<|im_start|>user
{Question} Reason step by step and output the final answer (A, B, C, or D) within \boxed{}.
<|im_end|>
<|im_start|>assistant

# B   Case Study

**Qwen2.5-Math-7B's Answer to the 18-th Question in AMC23**

**Question: Janet rolls a standard $6$-sided die $4$ times and keeps a running total of the numbers she rolls. What is the probability that at some point, her running total will equal $3$? The final answer can be written in the form $\frac{m}{n}$, where $m$ and $n$ are relatively prime positive integers. What is $m + n$? Response:** To solve this problem, we need to determine the probability that at some point during 4 rolls of a standard 6-sided die, the running total will equal 3. We can use dynamic programming to keep track of the probabilities of each possible running total after each roll. Here's the plan:

1. Initialize a list to keep track of the probability of each possible running total after each roll.
2. Start with the initial condition that the probability of a running total of 0 after 0 rolls is 1.
3. For each roll, update the probabilities of each possible running total by considering the outcomes of rolling a 1, 2, 3, 4, 5, or 6.
4. After each roll, check if the probability of a running total of 3 is non-zero.
5. The final answer will be the probability of a running total of 3 after 4 rolls. Let's implement this in Python:

...

(We omit the python code for readability.)

...

It looks like there was an issue with the implementation. The probability of the running total being exactly 3 after 4 rolls should not be zero, so let's correct the approach and re-implement the solution. Here's the corrected plan:

1. Initialize a list to keep track of the probability of each possible running total after each roll.
2. Start with the initial condition that the probability of a running total of 0 after 0 rolls is 1.
3. For each roll, update the probabilities of each possible running total by considering the outcomes of rolling a 1, 2, 3, 4, 5, or 6.
4. After each roll, check if the probability of a running total of 3 is non-zero.
5. The final answer will be the probability of a running total of 3 after 4 rolls.
Let's re-implement this in Python:

...

(We omit the python code for readability.)

...

It seems there is still an issue with the implementation. Let's manually check the logic and correct it step by step.

1. Initialize the probability list with the initial condition that the probability of a running total of 0 after 0 rolls is 1.
2. For each roll, update the probabilities of each possible running total by considering the outcomes of rolling a 1, 2, 3, 4, 5, or 6.
3. After each roll, check if the probability of a running total of 3 is non-zero.
4. The final answer will be the probability of a running total of 3 after 4 rolls. Let's manually simulate the process: - After 0 rolls: Probability of 0 is 1

...

The probability of a running total of 3 after 4 rolls is 4/1296, which simplifies to 1/324. Therefore, the final answer in the form of m/n is 1/324, and the sum m + n is 1 + 324 = 325. The final answer is $\boxed{325}$.

## C   Implementation Details about Semantic Clustering

We detail the implementation of semantic clustering in our EMPO. The semantic cluster process is shown as Algorithm 1. We also provide the simplified python code to show how we determine whether two model outputs are equivalent or not in mathematical reasoning (Algorithm 2) and free-form natural reasoning tasks (Algorithm 3).

---

**Algorithm 1:** Semantic Clustering

**Require** : question $q$, a group set of model response $\{o_2, \ldots, o_G\}$, verifier $\mathcal{V}$
**Initialize :** $C = \{o_1\}$
1  **for** $2 \leq i \leq G$ **do**
2     **for** $c \in C$ **do**
         // Random choose one element from $c$ for comparison
3        $o_c = c[0]$
         // Is the meaning of old sequence equivalent to new one?
4        **if** $\mathcal{V}(q, o_c, o_i) ==$ True **then**
            // Put into existing class
5           $c = c \cup \{o_i\}$ **break**
6        **end**
7     **end**
      // $o_i$ is semantically distinct, belongs to a novel cluster.
8     $C \leftarrow C \cup \{o_i\}$
9  **end**
**Return**  : $C$

---

**Algorithm 2:** Implementation of verifier for mathematical reasoning tasks.

```python
from math_verify import parse, verify

def are_equivalent(model_output_1, model_output_2)
    prediction_1 = parse(model_output_1)
    prediction_2 = parse(model_output_2)
    return verify(prediction_1, prediction_2)
```

---

## D   Additional Results on Llama3 Model Series

We conduct additional experiments to validate the efficacy of our EMPO on other model series beyond Qwen2.5. The results are shown in Table 4. Consistent with other concurrent practice, we are unable to implement R1-Zero-like training on the Llama series, i.e., directly initializing RL process from the Base model without SFT). Thus, we instead consider a semi-supervised learning approach by initializing from instruct-tuned model and enhance the reasoning capability with our EMPO. As shown in Table 4, when initialize from Llama3.2-3B-Instruct model, our EMPO can also substantially improve reasoning capability of instruct-tuned model which have undergone carefully-designed post-training.

Consistent with open-source community practices, we found that R1-Zero-like RL training can only be reproduced unsupervised on Qwen2.5 series Base models. In contrast, Llama3 series model still necessitate "cold-start", i.e., SFT, before RL. Specifically, in our experiments, the Qwen2.5 Base models demonstrated inherent answer consistency from the initial stages of EMPO training. However, Llama3 series Base models suffer severe inconsistency and fail to convergence during training. We hypothesize this divergence stems from Qwen2.5's pretraining strategy. As mentioned in the technical report [5], the pretrain data corpus are mixed with both web text and QA pairs generated by instruct-tuned Qwen2 models. This endows Qwen2.5 Base models with native instruction-following capabilities. Experimental evidence supports this hypothesis. As shown in Table 2, Qwen2.5 Base models successfully follow the instruction such as "put the final answer (A-P) within box" when answering multiple-choice questions from MMLU Pro and achieve an accuracy notably higher than random guess.

**Algorithm 3:** Implementation of verifier for natural reasoning tasks.

```python
verifier = AutoModelForCausalLM.from_pretrained(...)
tokenizer = AutoTokenizer.from_pretrained(...)

def are_equivalent(model_output_1, model_output_2,
    question, verifier)
    prediction_1 = parse(model_output_1)
    prediction_2 = parse(model_output_2)
    prompt = (
            f"User: ### Question: {question}\n\n"
            f"### Ground Truth Answer: {prediction_1}\n\n"
            f"### Student Answer: {prediction_2}\n\n"
            "For the above question, please verify if the
                student's answer is equivalent to the
                ground truth answer.\n"
            "Do not solve the question by yourself; just
                check if the student's answer is
                equivalent to the ground truth answer.\n"
            "If correct, output \"Final Decision: Yes\".
                If incorrect, output \"Final Decision:
                No\".\n"
            "Assistant: Final Decision: "
        )
    inputs = self.tokenizer(modified_prompt,
                    return_tensors="pt").to(self.model.device)
    input_ids = inputs.input_ids

    # inference for output logits
    with torch.inference_mode():
        outputs = self.model.forward(input_ids)
    logits = outputs.logits

    # get next output logits
    next_token_logits = logits[0, input_ids.shape[1] - 1,
        :]

    # get the token ID of "Yes" and "No"
    decision_tokens = self.tokenizer("Yes", "No")
    yes_id = decision_tokens.input_ids[0]
    no_id = decision_tokens.input_ids[1]

    # calculate probability
    probs = torch.softmax(next_token_logits, dim=0)
    yes_prob = probs[yes_id].item()
    no_prob = probs[no_id].item()

    return yes_prob > no_prob
```

Table 4: Accuracy on mathematical reasoning benchmarks.

| | Supervision | MATH | Minerva Math | OMNI | AIME24 | AMC23 | Avg. |
|---|---|---|---|---|---|---|---|
| *frontier model* | | | | | | | |
| Llama-3.1-70B-Instruct | $\{q, r, a\}$ | 64.6 | 35.3 | 31.9 | 16.7 | 30.1 | 35.7 |
| Eurus-2-7B-PRIME | $\{q, r, a\}$ | 79.2 | 38.6 | 42.1 | 26.7 | 57.8 | 48.9 |
| *1B model* | | | | | | | |
| Llama3.2-Instruct | None | 27.2 | 5.1 | 5.6 | 0.0 | 10.0 | 9.6 |
| Llama3.2-Instruct w/GRPO | $\{q, a\}$ | 29.8 | 3.7 | 6.4 | 0.0 | 12.5 | 10.5 |
| Llama3.2-Instruct w/EMPO | $\{q\}$ | 31.0 | 5.1 | 7.9 | 3.3 | 7.5 | 11.0 |
| *3B model* | | | | | | | |
| Llama3.2-Instruct | None | 46.2 | 19.1 | 15.3 | 3.3 | 20.0 | 20.8 |
| Llama3.2-Instruct w/GRPO | $\{q, a\}$ | 49.2 | 22.4 | 17.6 | 13.3 | 32.5 | 27.0 |
| Llama3.2-Instruct w/EMPO | $\{q\}$ | 49.8 | 20.2 | 18.4 | 13.3 | 30.0 | 26.3 |

# E    Additional Training Details

We provide a brief summary of our training recipes in Table 5. Besides, we have release the code in the supplementary materials which contained the full training configurations for re-implementation.

Table 5: A brief summary of training recipes of Qwen2.5 Base models.

| | 1.5B-Math | 7B-Math | 3B | 7B | 14B |
|---|---|---|---|---|---|
| Number of generations | 7 | 7 | 12 | 12 | 12 |
| Learning rate | 3e-7 | 3e-7 | 3e-7 | 3e-7 | 3e-7 |
| Max completion length | 2048 | 2048 | 1024 | 1024 | 768 |
| Batch size per GPU | 1 | 2 | 1 | 1 | 1 |

# F    Computational Cost of Semantic Clustering

Given the number of responses sampled per question $G$ (i.e., the group size) and the training dataset size $N$, the time complexity of the clustering process is $O(G^2 \times N)$. In mathematical reasoning tasks, semantic clustering is implemented by regular expressions which do not involve notable computational cost. For natural reasoning tasks, we rely on an additional compact small language model. To evaluate the additional computational overhead introduced by semantic clustering in EMPO, we conducted comparative analyses of EMPO and GRPO in terms of total training duration and GPU memory utilization. The results of mathematical reasoning and natural reasoning are shown in Table and Table 7, respectively. It is worthy to note that the 14B model experiments requires slightly less computational time than the 7B model. This is because, in our 14B experiments, we reduced the batch size and maximum response length from 2 and 1024 to 1 and 768, respectively, compared to the 3B and 7B configurations. This adjustment was made to fit the limited GPU memory of one single 8×A100 80G machine.

Table 6: Comparison of total runtime (measured as 8× A100 GPU hours) and storage cost (measured by max total GPU memory (GiB) utilization) between GRPO and EMPO. The GPU Memory semantic cluster process requires minimal computation and storage.

| | Qwen2.5-1.5B-Math | | Qwen2.5-7B-Math | |
|---|---|---|---|---|
| | GPU Hours | GPU Mem | GPU Hours | GPU Mem |
| GRPO | 11.2 | 240.4 | 8.5 | 501.3 |
| EMPO | 11.7 | 208.2 | 8.7 | 532.7 |

Table 7: Comparison of total runtime (measured as $8\times$ A100 GPU hours) and storage cost (measured by total GPU memory (GiB) utilization) between GRPO and EMPO. The GPU Memory semantic cluster process requires minimal computation and storage.

| | Qwen2.5-3B | | Qwen2.5-7B | | Qwen2.5-14B | |
|---|---|---|---|---|---|---|
| | GPU Hours | GPU Mem | GPU Hours | GPU Mem | GPU Hours | GPU Mem |
| GRPO | 9.5 | 274.8 | 12.4 | 508.6 | 11.0 | 588.2 |
| EMPO | 11.1 | 286.9 | 14.6 | 532.7 | 11.5 | 541.1 |

## G Details of Prompt Collection

For mathematical reasoning, we directly use 20,000 prompts randomly selected from Numina-Math-CoT. For free-form natural reasoning tasks, we adopt the prompts from Natural Reasoning[8] by filtering out the questions with over-long prompt, reference answer. Besides, we use the response length of Llama3.3-70B-Instruct as a difficulty estimation metric, and filter out overly difficult samples with response lengths exceeding 4096 tokens. The data collection python code is demonstrated as follow:

---

**Algorithm 4:** Python code of data filtering in a huggingface-like style.

---

```python
from datasets import load_dataset

dataset = load_dataset("facebook/Natural-Reasoning")

filtered_dataset = dataset.filter(
    lambda x: (
        # no answer
        len(x["reference_answer"]) > 0
        # over-long answer
        and len(x["reference_answer"]) < 129
        # overly difficult questions
        and len(x["llama_responses"]) < 4096
        # over-long prompt
        and len(x["question"]) < 512
        # proof-oriented
        and ("prove" not in x["question"].lower())
        and ("proof" not in x["question"].lower())
    )
)
```

---

## H Additional Results about Pass@k

We provide additional visualization pass@k results of models trained with EMPO. The results are shown as follow. As shown in Figure H, the Base model consistently catch up with RL trained models when k is large.

## I Discussion about Concurrent Works

Recent works explore self-supervised RL with various internal reward signals including majority voting (TTRL [15], SRT [46]), semantic coherence (our EMPO), token-level confidence score (Intuitor [47], One-shot Entropy Minimization [48]), and self-play (Absolute Zero [49]). TTRL and SRT generate pseudo label by majority voting, which is limited to close-end mathematical reasoning. Besides, TTRL directly conduct RL on test prompts (test-time training), while our EMPO keeps the strict separation between training and testing. Intuitor and One-shot EM use the token level softmax probability as confidence. By contrast, EMPO rewards coherent outcomes regardless the token-level probability. Besides, Absolute Zero further remove the dependency of prompts, which only relies on a compile environment only.

---

[8]https://huggingface.co/datasets/facebook/natural_reasoning

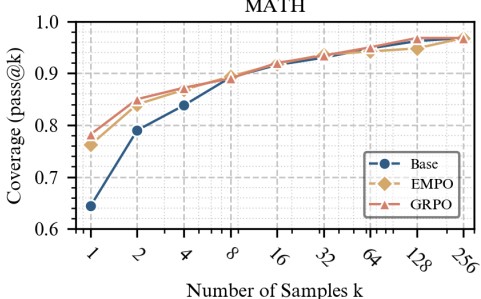
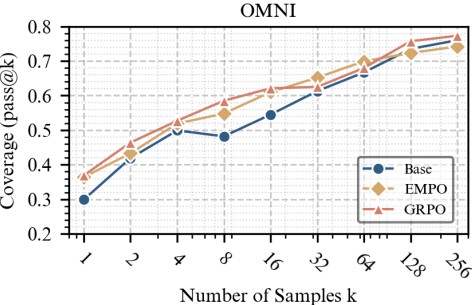

Figure 5: Trend of pass@k accuracy on Math test-set.

Figure 6: Trend of pass@k accuracy on OMNI test-set.

## J The Influence of Clustering Quality on the Performance of EMPO

In our mathematical reasoning experiments, semantic clustering is achieved solely through regular expression matching without introducing additional models. Due to the naturally structured response formats in mathematical tasks, regular expression could accurately determine answer equivalence, resulting in relatively high clustering quality.

However, in more general free-form natural reasoning tasks where model responses are free-form much more diverse (e.g., matrix, numbers, a few lines of sentences/codes...), the clustering quality can impact EMPO's effectiveness. For instance, in our more early practice, we tried DeBERTa (a bert-like model with 300M parameters trained by microsoft) for semantic clustering. Due to the poor quality of semantic clustering, our EMPO straggled to scale up and suffered from frequent reward hacking. Subsequently, by leveraging the general-verifier released by Tiger-Lab (a fine-tuned Qwen2.5-1.5B-Math model) for clustering, we successfully generalized EMPO to more general free-form reasoning tasks. Noted that even though this small language model undergoes supervised finetuning, it serves within our fully unsupervised framework as a fixed utility function for semantic comparison, rather than serving as a external supervisor for task-specific feedback. There are several fundamental difference between cluster model and the reward model used in supervised RL:

- The cluster model does not evaluate output correctness relative to input queries. It just provides pairwise comparisons between the model's own outputs. That is, it only provides binary answer about "whether these two answer is the same?" rather than "which answer is better?".

- The cluster model does not provide any guidance, such as gradient information or hints on how to refine the reasoning traces.

- Compared to reward model or human-verifier golden answers, it can be much easier to implement such a cluster model. For example, in mathematical reasoning tasks, only regular expressions are enough for clustering. In natural reasoning tasks, a finetuned Qwen2.5-1B model can provide high quality semantic cluster results.

Essentially, this is related to the non-identifiability problem in statistical inference [50]. The issue of non-identifiability arises because multiple, distinct underlying states (potential "truths," or more accurately, different reasoning pathways or different clusters of incorrect answers) could produce the same pattern of relational signals (i.e., the same semantic clustering results).

