# OpenReview forum: "Right Question is Already Half the Answer: Fully Unsupervised LLM Reasoning Incentivization"
_NeurIPS.cc/2025/Conference — NeurIPS 2025 spotlight_

### Official Review · Reviewer_shZN · 2025-06-28

**Clarity:** 2
**Significance:** 3
**Originality:** 3
**Rating:** 4
**Confidence:** 3

**Summary:**

Reasoning models are growing in prominence. However, training LLMs for reasoning often requires substantial supervised data. It remains unclear how to elicit reasoning in LLMs in an unsupervised manner. Here, the authors develop a new method (EMPO) to encourage models towards better reasoning in an unsupervised manner.

**Questions:**

The authors primarily report accuracy. Is this averaged over multiple seeds? It would be helpful to see error bars, if possible.

What are the error bars in Figure 3 showing?

The authors filter out by length: what happens if you then test the resulting EMPO-guided model on long prompts? We would want reasoning models to handle difficult tasks! I may have misinterpreted some of the prompt filtering.

**Ethical Concerns:**

["NO or VERY MINOR ethics concerns only"]

**Final Justification:**

The authors did address my concerns around error bars and plan to add a limitations. I am happy with the substantial amount of work they put through during revisions. However, I worry that some of the changes may require a more substantive rewrite on clarity that may be beyond what is "easy" to do during revisions, hence, my Borderline Accept. I think the direction of the work is promising and hope the authors continue!

**Limitations:**

Unfortunately, the authors did not include an explicit limitations section nor a negative societal impacts section. A limitations section is definitely needed. The authors are good about discussing the relative merits of their approach but could be more upfront on the current weaknesses of the method and experimental paradigm for testing thus far.

**Paper Formatting Concerns:**

The tables were hard to interpret (see above) and some of the Supplement (e.g., Section G) would benefit from a prose-polishing pass. But that did not impact my score.

**Quality:**

3

**Strengths And Weaknesses:**

The paper was strongly motivated. The authors do a good job of motivating that they take on cases without a clear "ground truth" answer -- many real problems/queries are like that, and we want our reasoning systems to be robust to handling them! To my understanding, EPMO is an original method (though I'm not too familiar with methods for incentivizing reasoning so may have missed something). The overall idea is clever, intuitive, and well-explained: all strengths to the paper.

I also thought the authors had a good discussion on the relative role of pre-training vs. reasoning elicitation.

With that said, I found the experimental results quite challenging to work through. The large tables were not easy to interpret. I appreciated that the authors included the amount of supervising information provided (that was helpful for assessing the methods), but it would have been helpful for more discussion and work on the presentation of results. (more noted in the Questions below).

If the experiment section were conveyed more clearly, I think this could be a very strong paper.

---

> ### Author Rebuttal · Authors · 2025-07-29
>
> We thank the reviewer for recognizing the original, intuitive and well-explained method and good discussion of the relationship between reasoning-oriented post-training and pre-training. We appreciate your constructive and actionable suggestions and address your concerns as follows. It is our duty to clearly present our work and avoid any confusion.
>
> - Q1: The authors report accuracy. Is it averaged over multiple seeds? It would be helpful to see error bars if possible.
>
> Thanks for raising concerns about error bars. Training LLMs with multiple seeds and reporting error bars can be extremely costly, thus many existing works do not have error bars [1-5]. For example, training a 7B LLMs on Numina-Math-20K for 3 epochs would cost more than 48 hours on 8*A100 GPUs. **While training large models with multiple seeds is indeed computationally prohibitive as noted, we share your concern about reproducibility.** We went to great lengths during the rebuttal period and **retrain** Qwen2.5-Math-7B on Numina-Math (1 epoch) for 3 independent runs (utilized all our available computational resources during the rebuttal). The randomness of results stem from different training dynamics (e.g., order of mini-batches, stochastic optimization) and also test-time random sampling (e.g., top-p, top-k). For each independent run, we report the average accuracy over 16 random sampling. The sampling parameters are temperature=0.6, top-p=0.95 and top-k=20. As shown below, the mean@16 accuracy averaged over 3 independent runs is relatively stable with small deviations across 3 independent runs.
>
> | Test set | 1st run   | 2nd run  | 3rd run  | average | standard deviation |
> | -------------- | ----- | ----- | ---- | ----- | ----- |
> | MATH | 79.57   | 74.32  | 75.90  | 76.60 | 2.20 |
> | MNIV | 46.07   | 29.18  | 31.80  | 35.68 | 7.42 |
> | OMNI | 40.53   | 38.37  | 39.26  | 39.39 | 0.89 |
> | AMC | 58.90   | 63.44  | 59.84  | 60.73 | 1.96 |
> | AIME | 18.96   | 24.16  | 22.71  | 21.94 | 2.19 |
>
> We will further clarify where the randomness come from and the details of evaluation to ensure reproducability. **Besides, we promise to open all the source code, training recipe, model checkpoints, training logs and data once accepted.**
>
> - Q2: What are the error bars in Figure 3 showing?
>
> In Figure 3, a Gaussian smoothing has been applied to the actual experimental curves using the Python Seaborn package. Due to the statistical fluctuations between mini-batches, the smoothed curves allow for a clearer presentation of the overall trends. Thus, the error bars are primarily for better visualization, which do not have significant practical meaning. We will further clarify this to avoid confusion in revision.
>
> - Q3: What happens if you then test the resulting EMPO-guided model on long prompts?
>
> This is a very important question about the generalizability of EMPO to more challenging, longer prompts. We fully agree with your opinion that difficult tasks would be more deserving to be handled with reasoning models. However, **please kindly note that Qwen2.5-Math models have limited max position embeddings due to its original model design**. Due to the maximum positional encoding length limit, the sum of the input prompt and response length must remain within **4096** tokens. Consequently, we filter out excessively long prompts to allow the model to generate more tokens for generating long reasoning traces, which is a choice consistent with common practice [6].
>
> To directly address your concerns, we conducted new experiments on OctoThinker-3B, which has a much longer maximum positional embeddings (131072). We used DeepMath-103K as the training set and set the maximum prompt length to a longer **4096 tokens (covering all problems)**. Under this setup, EMPO still demonstrates significant performance improvement, where the model's average response length exceeds 12K tokens. This demonstrates that the prompt length filtering in our main experiments was a practical choice due to model constraints, not a limitation of the EMPO framework.
>
> |  | Max prompt length   | Max generated tokens  | Accuracy  |
> | -------------- | ----- | ----- | ---- |
> | OctoThinker-3B-Base | 4*1024   | 12*1024  | 14.38  |
> | +EMPO | 4*1024   | 12*1024  | 31.41  |
>
>
> - Limitation and Social Impact
>
> Thanks for your kind reminder. **It is our duty to explicitly and well discuss the limitation and social impact.** This manuscript presents a new training algorithm that improves the reasoning capability by minimizing semantic entropy. By encouraging semantically coherent outputs, the model's reasoning capability can be boosted without external supervision. There exists several noteworthy limitation of EMPO. The unsupervised nature of EMPO removes the dependency of external supervision at the cost of potential reward hacking risks. Such limitation widely exists in unsupervised learning literature [7], but still noteworthy. Besides, as identified by [8], same as other RL for reasoning methods, EMPO may exhibit undesirable behaviors that stem from pretraining corpus of the Base model, e.g., hallucination, bias and so on. Entropy minimization may not only increase accuracy by encouraging consistency, but also make the bias more severe. Thus, more studies are neccessiated to mitigate such issues for large reasoning models. We promise to further elaborate the limitations and social impact in our revision.
>
> - Formatting concerns of tables and Appendix G.
>
> Thanks for your suggestion on formatting. We will 1) update mean@16 results 2) further explain how the metrics are calculated and where the randomness comes from in our experiments 3) add a explicit limitation and negative social impact section and 4) polish section G by replacing the python code with pesudo-code which is more suitable for acedamic writing. Besides, we promise to open all our code, model weights, training logs/recipes and data once accepted to make sure that the community can reproduce our results easily.
>
> [1]  rStar-Math: Small LLMs Can Master Math Reasoning with Self-Evolved Deep Thinking, ICLR'25 (142 citations)
>
> [2] Free Process Rewards without Process Labels, ICML'25 (55 citations)
>
> [3] Preference Optimization for Reasoning with Pseudo Feedback, ICLR'25 spotlight
>
> [4] Self-Consistency Preference Optimization, ICML'25
>
> [5] Advancing LLM Reasoning Generalists with Preference Trees, ICLR'25 (139 citations)
>
> [6] DeepMath-103K: A Large-Scale, Challenging, Decontaminated, and Verifiable Mathematical Dataset for Advancing Reasoning, Arxiv Preprint
>
> [7] The Entropy Enigma: Success and Failure of Entropy Minimization, ICML'24
>
> [8] Are Reasoning Models More Prone to Hallucination? Arxiv preprint

---

> > ### Comment · Reviewer_shZN · 2025-08-03
> > **Thank you for the detailed rebuttal**
> >
> > Dear Authors,
> >
> > Thank you for all the time you put in in the rebuttal! I especially admire you all retraining Qwen2.5-Math-7B. I am happy with the changes and have increased my score by a point; however, I do worry that the changes required to better clarify the results in the text warrants more work than may be easily ammenable during revisions. Hence, I do not envision moving beyond my (new heightened) "Borderline accept" score.
> >
> > I am very glad the authors do plan to add a limitations section and hope that is done in the first instance in future submissions!

---

> > > ### Author Response · Authors · 2025-08-04
> > > **Thanks for your support**
> > >
> > > Dear Reviewer,
> > >
> > > Thanks for your positive feedback and for raising your score.
> > >
> > > We understand your opinion and are actively working on completing all necessary clarifications to ensure the final manuscript is polished and clear. As promised, the limitations section will also be incorporated in the final revision.
> > >
> > > Best regards,
> > >
> > > Authors

---

### Official Review · Reviewer_Sarg · 2025-07-01

**Clarity:** 2
**Significance:** 2
**Originality:** 2
**Rating:** 4
**Confidence:** 3

**Summary:**

This paper introduces EMPO: a variant of GRPO rewarding models for minimizing response "semantic entropy". This is done first by clustering a set of responses $\{s_1,...,s_K\}$ into semantic clusters $\{c_1,...,c_C\}$ and then giving reward $r_i$ for $s_i$ as $p(c_{s_i})$ i.e. the probablity of the semantic cluster $c_{s_i}$. Experiments are conducted on math and free-form reasoning tasks.

**Questions:**

- Have you considered alternatives to final answer equivalence classes for reasoning? I think it would be very interesting to investigate the effect of different equivalence class schemes in more detail (this was briefly discussed in the final appendix section).

**Ethical Concerns:**

["NO or VERY MINOR ethics concerns only"]

**Limitations:**

The authors have **not** included a limitations and broader impacts section.

**Quality:**

2

**Strengths And Weaknesses:**

**Strengths:**

- The paper proposes a smart extension of entropy minimization ideas to LLM reasoning
- The paper is mostly clear and easy to follow
- The paper evaluates a good range of model sizes (3B, 7B, 14B) over multiple reasoning datasets

**Weaknesses:**

- The implementation details for how semantic entropy is measured are not present in the main body of the paper. For math problems, only the final answer is used to do clustering, resulting in an objective very similar to self-consistency rewards. Clustering for free-form problems is done using an open-source instruction model (which is fine but should be made clear much earlier).
- Because the final EMPO objective is so similar to self-consistency rewarding approaches, it would be nice to see a self-consistency rewarding baseline to compare to.

---

> ### Author Rebuttal · Authors · 2025-07-29
>
> We thank the reviewer for recognizing the smart idea, easy-to-follow writing and good evaluation range. We appreciate your positive review and address your concerns as follows.
>
> - W1: The implementation details for how semantic entropy should be made clear much earlier.
>
> Thanks for your carefully reading and suggestions. It is our duty to clearly present the implementation details. We will reorganize and provide a clear illustration of semantic clustering at the beginning of the method part according to your suggestion.
>
> - W2: It would be nice to see a self-consistency rewarding baseline to compare to.
>
> Thanks for mentioning the self-consistency rewarding baseline. According to your suggestion, we conduct additional experiments by training Qwen2.5-Math-1.5B on DeepMath-103K, and compare EMPO with 1) test-time majority voting and 2) self-consistency rewarding (SR). We report the accuracy average over 16 samplings, i.e., mean@16 of EMPO and self-consistency rewarding as the number of training steps increases.
>
> | Training step | 0   | 200  | 400  | 600  | 800  |
> | -------------- | ----- | ----- | ---- | ---- | ---- |
> | Self-consistency Rewarding   | 42.18  | 48.28  | 47.66 | 44.53  | 2.93 (collapse due to reward hacking) |
> | + Majority Voting (major@16)   | 56.21  | 58.48  | 57.82 | 52.22  | 2.52 |
> | EMPO   | 42.18  | 46.09 | 49.38  | 50.62 | 51.56 |
> | + Majority Voting (major@16)   | 56.12  | 56.98  | 61.50 | 59.86  | 63.23 |
>
> Compared to a self-consistency reward which is binary, EMPO's reward is "soften", offering better tolerance to reward noise. For example, assume the correct answer is "B". However, the LLM outputs the incorrect answer "A" 9 times and the correct answer "B" 7 times. In this scenario, self-consistency would yield an incorrect pseudo-label of "A" (based on the majority vote). The incorrect answer "A" would receive a reward of 1, while the correct answer "B" would receive a reward of 0, reinforcing an incorrect answer. In contrast, EMPO uses frequencies as rewards, assigning proximate rewards of 0.9 (for answer "A") and 0.7 (for answer "B"). Thus EMPO provides better tolerance to incorrect rewards, allowing the model to be less sensitive to the noisy model outputs. Noted that there exists several concurrent works utilized self-consistency rewarding [1-4], we primarily follow [1] to implement the self-rewarding with GRPO. We will add these results in our revision and further discuss the distinction between EMPO and [1-4].
>
> - Q1: Have you considered alternatives to final answer equivalence classes for reasoning? It would be very interesting to investigate the effect of different equivalence class schemes in more detail.
>
> You raised a very interesting question! Currently, we use regular expressions or another language model to determine the pairwise equivalence between answers. Expanding these equivalence classes may be necessary for diverse tasks such as retrieval and long-text generation. 1) For tasks like retrieval, text embedding similarity can be used to define equivalence classes. 2) For long-text generation tasks, previous work [5] has utilized external models to first construct factual questions based on the generated content. The model is then prompted to answer these questions based on its generated text, and the semantic entropy calculated individually for each question is aggregated to determine the factual reliability of the entire passage. In this circumstance, two generations are equivalent if and only if all the factual questions yield the same answers. By expanding the equivalence class, EMPO will have the potential to be adapted for those long-text generation tasks that require guaranteed factual correctness.
>
> Thanks for raising such an inspiring question. We will elaborate these future directions in our revision.
>
> - Limitation and Social Impact
>
> Thanks for your kind reminder. **It is our duty to explicitly and well discuss the limitation and social impact.** This manuscript presents a new training algorithm that improves the reasoning capability by minimizing semantic entropy. By encouraging semantically coherent outputs, the model's reasoning capability can be boosted without any external supervision. There exists several noteworthy limitation of EMPO. The unsupervised nature of EMPO removes the dependency of external supervision at the cost of potential reward hacking risks. That is, the model may be highly overconfident on its prediction regardless correctness. Besides, as identified by [6], similar to other RL for reasoning methods, EMPO may exhibit undesirable behaviors that stem from pretraining corpus of the Base model, e.g., hallucination, bias and so on. Entropy minimization may not only increase accuracy by encouraging consistency, but also make the bias more severe. Thus, more studies are necessitated to mitigate such issues for large reasoning models. We will add these discussion to our revision.
>
> [1] Preference Optimization for Reasoning with Pseudo Feedback, ICLR'25 spotlight
>
> [2] Self-Consistency of the Internal Reward Models Improves Self-Rewarding Language Models, Arxiv preprint
>
> [3] CREAM: Consistency Regularized Self-Rewarding Language Models, ICLR'25
>
> [4] Self-Consistency Preference Optimization, ICML'25
>
> [5] Detecting hallucinations in large language models using semantic entropy, Nature'24
>
> [6] Are Reasoning Models More Prone to Hallucination? Arxiv preprint

---

> > ### Comment · Reviewer_Sarg · 2025-08-04
> >
> > Thank you for your response, I will keep my positive score.

---

> > > ### Author Response · Authors · 2025-08-05
> > > **Thanks for your support**
> > >
> > > Thank you for the time and effort you spent on reviewing our work. We truly value the opportunity to have this discussion with you. Feel free to reach out if there are any questions or insights you would like to further discuss with us.

---

### Official Review · Reviewer_bd1c · 2025-07-03

**Clarity:** 3
**Significance:** 3
**Originality:** 3
**Rating:** 4
**Confidence:** 4

**Summary:**

This paper introduces Entropy Minimized Policy Optimization, a novel reinforcement learning algorithm aimed at enhancing the reasoning capabilities of large language models without any supervision. Existing methods rely on supervised fine-tuning or external reward signals; EMPO instead minimizes semantic entropy over multiple generations from unlabeled prompts. The authors cluster semantically equivalent outputs and assign higher rewards to more confident clusters, training the model to favor consistent reasoning traces. Empirical results show that EMPO significantly improves performance on both mathematical reasoning (e.g., MATH, Minerva) and free-form reasoning benchmarks (e.g., MMLU-Pro, GPQA), often matching or surpassing supervised approaches.

**Questions:**

- The paper introduces dual thresholds to exclude examples with overly low or high entropy during training. Could the authors explain how these thresholds were selected in practice?
- In the natural reasoning tasks, the authors utilize a verifier model to determine whether two outputs are semantically equivalent, using a prompt such as "Please verify if the student’s answer is equivalent to the ground truth answer." This raises some questions about the underlying assumptions of the task setup. Specifically, are the natural reasoning prompts used in this work associated with deterministic, canonical answers that can reasonably serve as “ground truth” for such verification? If so, this setup is more straightforward. However, in more open-ended tasks, such as translation, summarization, or rewriting, there may not exist a single correct answer, and the notion of equivalence becomes less well-defined. In such cases, relying on a single model’s binary judgment of equivalence may introduce noise or bias into the clustering process. It would be helpful if the authors could clarify the scope of applicability of their current semantic clustering approach and discuss whether any adaptations would be needed for tasks with inherently diverse valid outputs.

**Ethical Concerns:**

["NO or VERY MINOR ethics concerns only"]

**Final Justification:**

I confirm that the authors have addressed my concerns and questions.

**Limitations:**

Yes

**Quality:**

3

**Strengths And Weaknesses:**

**Strengths**
- The paper is well-structured with clear motivation.
- The paper conduct extensive experiments on both mathematical reasoning and free-form reasoning benchmarks.
- The Section 5 discussion of the role of unsupervised learning in LLM reasoning is promising.

**Weaknesses**
- The authors acknowledge the risk of reward hacking under unsupervised objectives and propose a dual entropy thresholding strategy to filter out prompts that are either too uncertain or already overconfident. While this is a reasonable design, the paper does not present any ablation study or sensitivity analysis to empirically justify this choice. In particular, there is no evidence on how different threshold values affect training stability or final model performance. Given the critical role this filtering plays in avoiding trivial solutions and stabilizing training, a deeper empirical exploration would be valuable.
- Given the rapid pace of development in this area, a number of concurrent works have emerged that also explore label-free RL for language models. While it is understood that these works are likely developed in parallel, a brief discussion comparing EMPO to these methods could strengthen the paper. Adding a discussion section, perhaps in the appendix, highlighting the distinctions in methodology, assumptions, or application scope would help readers better understand EMPO’s specific contributions in the broader context of this growing field.
- Since EMPO relies entirely on unlabeled data, the quality and diversity of the training prompts may play a significant role in its effectiveness. It would be helpful to understand how the method performs under varying amounts of data or different prompt selection strategies. For example, does performance scale linearly with more data, or is it sensitive to prompt difficulty or domain distribution? Including a brief analysis or reflection on this point would enhance understanding of the method’s robustness and practical applicability.

---

> ### Author Rebuttal · Authors · 2025-07-29
>
> We sincerely thank the reviewer for your valuable comments and appreciate your recognition of the clear motivation, extensive experiments and promising discussion. We provide detailed responses to address your concerns.
>
> - W1: The paper does not present any ablation study or sensitivity analysis to empirically justify the choice of entropy thresholds.
>
> Thanks for raising this comment. We conduct two additional experiments involving Qwen2.5-Math-1.5B, Qwen2.5-7B on the very recent DeepMath-103K dataset (released at 2025.04) to ablate entropy thresholding. Since it is predictable that when we remove $\delta_{low}$, the peak performance will not change too much but the training will become slower (which is also mentioned by reviewer 3qKR). Thus, due to the high computational cost of LLM ablation studies, we preliminarily ablated the entropy upper bound ($\delta_{high}$). We report the ratio of filtered prompts, reward noise (mean absolute error between EMPO's unsupervised reward and GRPO's correctness reward), the final accuracy on AMC23 testset (averaged by 16 random sampling, i.e., mean@16).
>
> Qwen2.5-Math-1.5B
> | $\delta_{high}$ | Reward Noise   | Accuracy  |
> | ----- | ----- | ----- |
> | 0.0   | 0.625  | 47.66 (early stopped before reward hacking) |
> | 0.25   | 0.422    |   46.56    |
> | 0.5       | 0.228 | 51.56 |
>
> Qwen2.5-7B
> | $\delta_{high}$ | Reward Noise   | Accuracy  |
> | ----- | ----- | ----- |
> | 0.0   | 0.413  | 39.69 |
> | 0.25   | 0.346    |   43.59    |
> | 0.5       | 0.216 | 48.12 |
>
> As shown above, the thresholding effectively reduces the noise of reward signals and improves the performance as we expected.
>
> - W2: A brief discussion comparing EMPO to other concurrent works could strengthen the paper.
>
> We sincerely appreciate your kind understanding and suggestions.
>
> We extend our discussion in line 72-87 to further explain the distinction between EMPO and other concurrent works. Recent works explore self-supervised RL with various internal reward signals including majority voting (TTRL [1], SRT [2]), semantic coherence (our EMPO), token-level confidence score (Intuitor [3], One-shot Entropy Minimization [4]), and self-play (Absolute Zero [5]). TTRL and SRT generate pseudo label by majority voting, which is limited to close-end mathematical reasoning. Besides, TTRL directly conduct RL on test prompts (test-time training), while our EMPO keeps the strict separation between training and testing. Intuitor and One-shot EM use the token level softmax probability as confidence. By contrast, EMPO rewards coherent outcomes regardless the token-level probability. Besides, Absolute Zero further remove the dependency of prompts, which only relies on a compile environment only. **We will definitely add this discussion in our revision. If there are any representative papers we missed, please let us know.**
>
> - W3: How EMPO performs under varying amounts of data or different prompt selection strategies? Does performance scale linearly with more data, or is it sensitive to prompt difficulty or domain distribution? A brief analysis or reflection on this point would help.
>
> Very insightful question! We appreciate your carefully reading and in-depth suggestions. To investigate the scalibility and sensitivity of EMPO, we conduct additional experiments and train OctoThinker-3B and Qwen2.5-7B with EMPO on the very recent, large-scale DeepMath-103K (more than 5x larger than the training set we used in paper). The test accuracy (mean@16) on AMC'23 of varying training samples are shown as follows
>
> | Training step | 0   | 100  | 200 | 300 | 400 | 500 |
> | ----- | ----- | ----- | ----- | ----- | ----- | ----- |
> | Training samples   | 0  | 6.4K | 12.8K  | 19.2K | 25.6K  | 32.0K  |
> | Accuracy of Qwen2.5-7B   | 27.81  | 35.94 | 38.75  | 39.69 | 44.69 | 48.12 |
> | Accuracy of OctoThinker-3B   | 14.37  | 26.87 | 31.41  | 24.82 | 12.06 | 0.00 |
>
> As shown in above, EMPO consistently scales up the performance of Qwen2.5-7B on larger and more diverse training dataset. The model accuracy on AMC23 testset steadily increases along with the amount of data.
> However, for OctoThinker-3B [6] which is trained from weaker base model (Llama3-3B), the performance improves at the first 200 steps and reaches a plateau. Scaling up performance with long-term (supervised or unsupervised) RL may still be a challenging systematic engineer, including entropy controlling, data diversity, training efficiency and AI infra [8]. We promise to make this point transparent in our revision.
>
> - Q1: Could the authors explain how these thresholds were selected in practice?
>
> We suggest two ways to determine the thresholds.
>
> 1) We can employ EMPO with a smaller LLM (e.g., qwen2.5-math-1.5B) and **monitor the training-time semantic entropy** to detect "hacking". When hacking occurs, the semantic entropy typically drops rapidly to nearly 0 within <20 steps. Through such a pre-experiment, we can then choose a suitable threshold for training larger LLMs (>=7B). To illustrate this, we present the change in semantic entropy during hacking while training Qwen2.5-Math-1.5B on DeepMath-103K without thresholding.
> | Training step |1008  | 1012 | 1016 | 1020 | 1024 |
> | ----- | ----- | ----- | ----- | ----- | ----- |
> | Semantic entropy   | 2.995  | 2.367 | 1.580  | 1.136 | 0.343  |
>
> 2) Alternatively, we can **inference offlinely on the unlabel dataset for multiple times and determine the thresholds according to the average semantic entropy**. For example, when training Qwen2.5-7B on DeepMath, semantic entropy has a strong positive correlation with reward noise. Thus we can estimate the reward noise level by semantic entropy, and then choose suitable thresholdings.
>
>
> | $\delta_{high}$ | Reward Noise   | Semantic entropy  |
> | ----- | ----- | ----- |
> | 0.0   | 0.413  | 3.626 |
> | 0.25   | 0.346    |   3.575    |
> | 0.5       | 0.219 | 2.305 |
>
> Besides, in our experiments, setting the entropy upper threshold to 0.5 * maximum entropy (i.e., log K, where K is the number of semantic clusters) consistently prevented hacking.
>
> - Q2: Are the natural reasoning prompts associated with deterministic, canonical answers that can reasonably serve as “ground truth” for such verification? In more open-ended tasks there may not exist a single correct answer...It would be helpful to clarify the scope of applicability of current semantic clustering approach and discuss whether any adaptations would be needed for tasks with inherently diverse valid outputs.
>
> We sincerely thank for your carefully reading and in-depth comments. Your question led us to think a lot. EMPO's semantic entropy minimization objective is primarily designed to address "hallucinated" false, nonsensical or fabricated answers. Thus such an objective may not benefit tasks that necessitate divergent thinking like creative writing. However, for other tasks that require ensuring factual correctness, such as rewriting, translation, and summarization, our EMPO still has the potential to be adapted. For example, in long-text generation tasks, previous work [7] has utilized external models to first construct several factual questions based on the generated content. The model is then prompted to answer these questions based on generated text, and the semantic entropy calculated individually for each question is aggregated to determine the factual reliability of the entire passage. By leveraging such a **divide-and-conquer** approach, EMPO will have the potential to be adapted for those long-text generation tasks that require guaranteed factual correctness.
>
> **In the revision, we will define the current scope for convergent reasoning and explicitly detail these powerful adaptations for complex, diverse-output tasks as a promising direction for future work.**
>
> [1] TTRL: Test-Time Reinforcement Learning, Arxiv Preprint
>
> [2] Can Large Reasoning Models Self-Train?, Arxiv Preprint
>
> [3] Learning to Reason without External Rewards, Arxiv Preprint
>
> [4] One-shot Entropy Minimization, Arxiv Preprint
>
> [5] Absolute Zero: Reinforced Self-play Reasoning with Zero Data, Arxiv Preprint
>
> [6] OctoThinker: Mid-training Incentivizes Reinforcement Learning Scaling, Arxiv Preprint
>
> [7] Detecting hallucinations in large language models using semantic entropy, Nature'24
>
> [8] Prorl: Prolonged reinforcement learning expands reasoning boundaries in large language models, Arxiv Preprint

---

> > ### Comment · Reviewer_bd1c · 2025-08-04
> >
> > Thanks for your clarification. I keep my positive score for acceptance. Please include the additional results in your revisions. Good luck!

---

> > > ### Author Response · Authors · 2025-08-05
> > > **Thank you for your support**
> > >
> > > Thank you for your support. If there are any further insights or questions you would like to discuss with us, please feel free to reach out. We appreciate your positive assessment and time in reviewing our paper.

---

### Official Review · Reviewer_3qKR · 2025-07-05

**Clarity:** 3
**Significance:** 3
**Originality:** 4
**Rating:** 5
**Confidence:** 3

**Summary:**

This paper studies a new unsupervised learning paradigm for RL training of LLM reasoning and proposes Entropy Minimized Policy Optimization (EMPO) to elicit LLM reasoning performance via unsupervised learning without the need for ground truth labels. They show that when trained on mathematical reasoning and free-form natural reasoning tasks, RMPO achieves the same or even better performance than RL methods like GRPO and it does not need a manually labeled reward or even generated pseudo-reward from LLM (for example, by majority vote).

On a high level, I think this new paradigm is interesting and worth investigating, and has potential for future research. Although I think this research is in the initial stage of unsupervised training for LLM on reasoning tasks, I lean toward accepting this submission because of its novelty and the soundness of the experiments.

**Questions:**

See the strengths and weaknesses section.

**Ethical Concerns:**

["NO or VERY MINOR ethics concerns only"]

**Final Justification:**

Thanks for your detailed rebuttal, and I really appreciate the experiments on entropy thresholding.

And also, you addressed my concerns about some experiment details such as the prompt difference for different methods.

The results on the DeepMath-103K are promising and I really like it.

I believe your response greatly addresses my concerns and questions and will keep my score.

**Limitations:**

See the strengths and weaknesses section.

**Paper Formatting Concerns:**

/

**Quality:**

3

**Strengths And Weaknesses:**

Strengths:

1. I think the new paradigm this paper proposes about unsupervised training for LLM reasoning is novel and has huge potential, especially when we need a huge amount of data, and labeling these questions is hard, if not impossible. This is also suitable for the reasoning tasks that do not have ground-truth labels, such as some general open-form tasks.

2. The experiment results look good, and the performance of EMPO can match or even outperform the RL methods like GRPO, or other methods like SFT. I think this is surprising to me.

3. The discussion part is pretty clear, I think. When I read sections 1 to 4, I was thinking: does the pass@k rate improve when k is large? And does the total solvable set increase compared to the base model and supervised-RL-trained models? And they answer my questions in section 5, as they know what I want to read.

Weaknesses and Questions:

1. About the formulation 7: They have an entropy filtering applied to the questions that are trained in EMPO. I think their explanation is good, but I have two questions. (a). How many prompts do they filter from this entropy filter? (b). Could you show some experimental results about what happens if we remove the \delta_low or \delta_high (I think we will be more interested in the results when removing \delta_high as it is predictable that when we remove \delta_low, the peak performance will not change too much but the training will become slower).

2. About the experiment details: It seems to me that you do SFT and GRPO on a different training dataset. (not sure whether I understand it correctly. Please correct me if I am wrong) Why is that? Moreover, you mentioned you filter a part of the data based on the response length of Llama3.3-70B-Instruct? Did you do the same step for SFT and GRPO? I think you should make a clearer comparison of the difference between the EMPO and GRPO implementations.

3. I think the experiments do not show the most advantageous case of EMPO. I think the best case will be: there is a huge dataset with questions only, and labeling them is impossible, or the labels are messy. Will EMPO achieve a good performance on such a dataset?

4. Since EMPO does not rely on the ground-truth labels, there are no such signals. Then, it will be even more important to select a training dataset with moderate difficulty. How will EMPO perform when the dataset is very very hard? For example, if most answers generated are incorrect (but the incorrect answers are diverse so it still passes the entropy filtering), how will the model perform?

---

> ### Author Rebuttal · Authors · 2025-07-29
>
> We appreciate your thoughtful and thorough review and recognition of the interesting and novel paradigm. We provide detailed responses to the constructive comments.
> - Q1: How many prompts are filtered out by entropy thresholding? Could you show experimental results when we remove $\delta_{low}$ or $\delta_{high}$?
>
> According to your suggestions, we conduct additional experiments to ablate the entropy thresholding. We train Qwen2.5-7B with $\delta_{low}=0$ and $\delta_{high}$ ranges from {0.0, 0.25, 0.5}. The ratio of filtered prompts, mean absolute error between EMPO's unsupervised reward and GRPO's correctness reward (i.e., the noise of EMPO's reward signal), the final accuracy (average over 16 random sampling, i.e., mean@16) on AMC23 testset are as follow
> | $\delta_{high}$ | Filter ratio   | Reward noise  | Accuracy  |
> | -------------- | ----- | ----- | ---- |
> | $0.0$   | $0.000$  | $0.413$  |  $39.69$    |
> | $0.25$   | $0.503$     |   $0.346$    |  $45.59$    |
> | $0.5$            | $0.613$ | $0.219$ |   $48.12$   |
>
> As shown above, the thresholding effectively reduces the noise of reward signals and improves the final performance. We will add these results to our revision.
>
> - Q2: Why do SFT and GRPO use different train sets? Did the authors do the same data filter step for both SFT and GRPO? A clearer comparison of the difference between EMPO and GRPO implementations should be provided.
>
> Please kindly note that the Numina-Math and Natural-Reasoning training sets are labeled with both ground-truth and chain-of-thought reasoning traces. We use the **same** train prompts for SFT, GRPO and EMPO in our experiment. The only difference is the form of supervision. SFT and GRPO use CoT and golden final answer labels respectively. Our EMPO only needs prompts. Besides, for the natural reasoning task, **we do data filtering for both SFT and RL methods (EMPO, GRPO) to make sure they are trained on the same subset**.
>
> The implementation difference between GRPO and EMPO lies in the calculation of reward signal. While GRPO relies on groundtruth, our EMPO uses the frequency of meaning clusters as internal reward. Besides, as suggested by DAPO [1], we also remove the KL penalty from the reward, which benefits training efficiency.
>
> We thank the reviewer for these suggestions and will further clairify the implementation details in revision.
>
> - Q3: Will EMPO achieve good performance on a huge dataset with questions only?
>
> You raise a very interesting comment. Unlabeled questions in the wild arise freely upon deploying LLMs in the open world, of which the amount should be much larger than that of labeled ones. Since reasoning-oriented reinforcement learning is an emerging and rapidly developing field, to our best knowledge, there are no such benchmarks yet. To investigate the scalibility of EMPO, we train Qwen2.5-7B on the very recent DeepMath-103K (released on 2025.04, **5 times larger** than the training set we used in the manuscript). The model accuracy on AMC23 testset steadily increases along with the scale of training samples.
>
> | Training step | 0   | 100  | 200  |  300   | 400  | 500  |
> | -------------- | ----- | ----- | ---- | ----- | ----- | ---- |
> | Total samples   | 0K  | 6.4K  |  12.8K    | 19.2K  | 25.6K  |  32K    |
> | Accuracy   | 27.81     |   35.94    |  38.75    | 39.87     |   44.69    |  48.12    |
>
> - Q4: How will EMPO perform when the dataset is very very hard?
>
> Very insightful question! To investigate this problem, we run OctoThinker-3B (a weaker LLM trained from Llama3-3B-Base) on diverse large-scale DeepMath-103K dataset. In this circumstance, the reward signal is expected to be rather noisy. The model accuracy improves at the beginning, but finally falls into a trivial solution.
>
> | Training step | 0   | 100  | 200  |  300   | 400  | 500  |
> | -------------- | ----- | ----- | ---- | ----- | ----- | ---- |
> | Accuracy   | 14.37     |   26.87    |  31.41    | 24.82     |   12.06    |  0.00    |
>
> Thus, if there are too many incorrect answers pass the entropy filtering, the unsupervised learning may become biased and ultimately collapses. This limitation widely exists in unsupervised learning field [2], but still noteworthy! Detecting incorrect but highly confident model answers would be a promising research direction. We promise to transparently discuss this failure mode in the Limitations section in our revision.
>
> [1] DAPO: An Open-Source LLM Reinforcement Learning System at Scale, Arxiv Preprint
>
> [2] The Entropy Enigma: Success and Failure of Entropy Minimization, ICML'24

---

> > ### Author Response · Authors · 2025-08-09
> > **A gentle reminder**
> >
> > Dear Reviewer 3qKR,
> >
> > Thank you for your positive accessment and valuable feedback. In our rebuttal, we have provided a detailed response to your questions. As the discussion period ends in a few hours, we would be happy to discuss any follow-up questions you may have. Thank you again for your time and effort!
> >
> > Best regards,
> >
> > Authors

---

### Note · Authors · 2025-08-11

Dear PCs, SACs, ACs, and Reviewers,

Thank you for your valuable feedback and insightful reviews. We are deeply encouraged that all reviewers unanimously gave positive reviews of our work. Reviewers found the proposed paradigm for LLM reasoning to be **novel, interesting, and holding great potential** (Reviewer 3qKR, shZN), and **a smart extension of entropy minimization ideas** (Reviewer Sarg). The paper was recognized for its **strong and clear motivation** (Reviewer bd1c, shZN), with an **intuitive and well-explained** method (Reviewer shZN). The experiments were described as **sound and extensive**, with a **good evaluation range** (Reviewer 3qKR, bd1c, Sarg).

In the rebuttal, we have addressed the following raised concerns:

- **Ablation Studies**. We conducted additional experiments to provide ablation studies on the entropy thresholding mechanism, justifying our design choices as requested by Reviewers 3qKR and bd1c.

- **Various Datasets**. To address questions about performance on difficult datasets and data amount (Reviewer 3qKR, bd1c), we provided new results training on the larger and more recent DeepMath-103K dataset.

- **Baseline Comparisons**: As suggested by Reviewer Sarg, we added a direct comparison with a self-consistency rewarding baseline, demonstrating EMPO's advantages in handling noisy model outputs.

- **Discussion about Concurrent Work**. We added a detailed discussion comparing EMPO's methodology and contributions to other recent concurrent works in label-free RL, as suggested by Reviewer bd1c.

- **Reproducibility and Clarity**. In response to Reviewer shZN's concerns, we retained the Qwen2.5-Math-7B model for 3 independent runs to provide standard deviations. We also clarified the visualization in Figure 3.

- **Scope and Limitations**. We clarified the applicability of our semantic clustering approach to more open-ended tasks and discussed potential adaptations (Reviewer bd1c). We also drafted a detailed limitations and social impact section to be included in our revision.

We believe these clarifications and additional results strengthen our paper and thoroughly address the reviewers' concerns. We are grateful for the time and effort you have dedicated to our submission.

Best regards,

Authors

---

### Decision · Program_Chairs · 2025-09-17

**Decision:**

Accept (spotlight)

**Comment:**

This paper introduces Entropy Minimized Policy Optimization (EMPO), an unsupervised approach for improving LLM reasoning. The main idea is to boost a model's reasoning skills without any labeled data by rewarding it for generating semantically consistent answers to unlabeled questions. The work is well-motivated, and the method itself is quite intuitive. Its core strength lies in its novelty and potential, offering a new direction for training models on tasks where supervision is hard to come by, and the experimental results are solid, showing that EMPO can match or even beat supervised methods. The initial reviews did bring up a few good points for improvements, such as more ablation studies on the entropy filtering mechanism, a clearer comparison to self-consistency baselines, and further discussion on how the method handles larger or more difficult datasets. The authors thoroughly addressed these concerns in their rebuttal with new experiments and clarifications, which strengthened the paper. I am happy to recommend acceptance.